# UniFolding: Towards Sample-efficient, Scalable, and Generalizable Robotic Garment Folding

**Han Xue[1,2,*], Yutong Li[1,*], Wenqiang Xu[1], Huanyu Li[1], Dongzhe Zheng[3], Cewu Lu[1,†]**

[1]Shanghai Jiao Tong University
[2]Shanghai AI Laboratory, Pujiang Lab
[3]University of New Hampshire

**Abstract:** This paper explores the development of UniFolding, a sample-efficient, scalable, and generalizable robotic system for unfolding and folding various garments. UniFolding employs the proposed UFONet neural network to integrate unfolding and folding decisions into a single policy model that is adaptable to different garment types and states. The design of UniFolding is based on a garment's partial point cloud, which aids in generalization and reduces sensitivity to variations in texture and shape. The training pipeline prioritizes low-cost, sample-efficient data collection. Training data is collected via a human-centric process with offline and online stages. The offline stage involves human unfolding and folding actions via Virtual Reality, while the online stage utilizes human-in-the-loop learning to fine-tune the model in a real-world setting. The system is tested on two garment types: long-sleeve and short-sleeve shirts. Performance is evaluated on 20 shirts with significant variations in textures, shapes, and materials. More experiments and videos can be found in the supplementary materials and on the website: `https://unifolding.robotflow.ai`.

**Keywords:** Deformable Object Manipulation, Bimanual Manipulation, Garment Folding

## 1 Introduction

Garment manipulation has been a long-standing task in the robotics community, with the potential to automate this process and enhance the quality of life by reducing human labor. However, despite recent advancements in learning methods for garment unfolding and folding [1, 2, 3, 4], they still struggle to efficiently handle the wide variety of garments within the same category that differs in shapes, sizes, textures, and materials. This limitation hampers the applicability of these methods in real-world applications.

Garments have unique properties that challenge large-scale data collection, such as the high-dimensional state space, self-occlusion, and complex dynamics [5]. Recently, there have been two lines of learning-based methods for garment manipulation. One line of works [2, 6] directly collects demonstration data from one or two garments in the real world without simulation, a process that challenges scalability and the achievement of high generalization capacity. Another line of works [7, 3, 8, 9, 1] utilize simulation data for training, which requires a large number of samples [7, 3] or complete cloth mesh [1, 8, 9] for policy learning, but it is infeasible in real world. Besides, these methods suffer from sim2real gaps because many garment states and dynamics cannot easily be covered in simulators [10, 11, 12]. Thus, it is desirable to adopt real-world data for fine-tuning. However, efficiently utilizing and annotating real-world data at low cost is a significant challenge.

In this paper, we propose a novel robot manipulation system **UniFolding** for generalizable garment folding. It leverages an end-to-end neural network **UFONet** to make action decisions. Given a garment in a crumpled state, the system first unfolds the garment through *fling* actions, then folds the garment through pick-and-place actions (see Fig. 1). UFONet takes partial point cloud as input,

---

\* These authors contributed equally to this work.     † Cewu Lu is the corresponding author.

7th Conference on Robot Learning (CoRL 2023), Atlanta, USA.

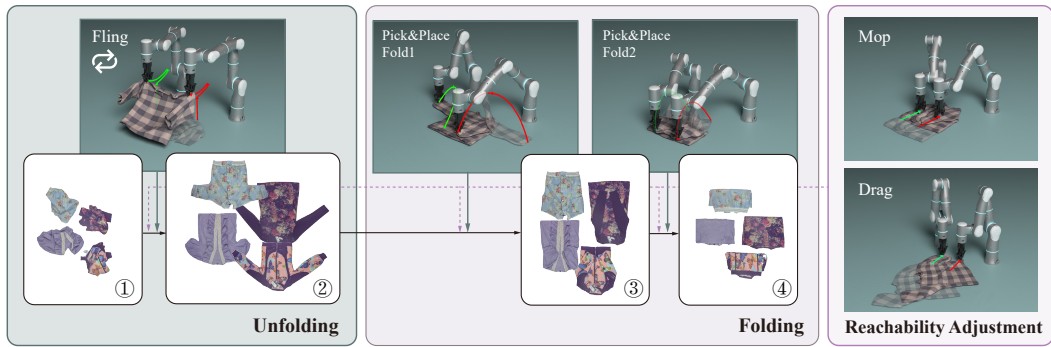

Figure 1: The manipulation pipeline of UniFolding system. It contains two stages to fully fold a garment from an initial crumpled state, namely *Unfolding* and *Folding*.

which is less sensitive to the texture or shape diversity than the 2D-based solutions [1, 2]. Besides, UFONet unifies the unfolding and folding policy into one model, and it can handle corner cases where simple heuristic folding rules (*i.e.*, keypoint detector [1] or template matching [2]) may fail.

The training pipeline of UFONet prioritizes low-cost and sample-efficient data collection. We devise a human-centric pipeline (see Fig. 2) which consists of offline data collection in simulation with human demonstration and online data collection in the real world. In the offline data collection phase, we collect human demonstration data for unfolding and folding tasks through a Virtual Reality interface [13] in a fast and low-cost manner. By leveraging human priors from the demonstrations, we can simplify the dense action space into a ranking problem with a sparse set of keypoint candidates. This substantially reduces exploration time in both simulation and the real world. After obtaining an initial policy from offline supervised learning, we perform self-supervised learning in simulation for unfolding tasks. In the online data collection phase, we adopt a human-in-the-loop learning approach to fine-tune the policy in the real world. Experiments show that only a few annotations in the online data collection phase can largely improve the unfolding performance.

To evaluate the folding system, we conducted experiments on long-sleeve shirts and short-sleeve shirts with high textures, shapes, and material variance. We measure our approach's unfolding and folding performance in Sec. 5. We summarize our contribution as follows:

- We propose a novel robotic folding system, **UniFolding**, that can support the complete garment folding pipeline including *unfolding* and *folding*.
- We propose **UFONet**, an end-to-end policy model along with a training pipeline for efficient policy training in the real world.
- We conduct extensive real robot experiments on 10 *unseen* long-sleeve shirts and 10 *unseen* short-sleeve T-shirts to demonstrate the generalization ability and robustness of our system.

## 2 Related Works

**Learning-based Cloth Unfolding.** Most learning-based methods for cloth unfolding [7, 3, 8, 9, 1, 4] rely on real-time cloth simulators (*i.e.*, Pyflex [10]) for data collection. However, many complex garment states, materials, and dynamics can not be accurately modeled by these PBD-based [14] simulators [10, 11, 12]. Thus, the sim2real gaps are the key obstacles for these methods to achieve better generalization ability in real-world applications.

Unfortunately, large-scale data collection for cloth manipulation in the real world is difficult for previous methods [15]. Some methods rely on complete cloth mesh to calculate rewards [1] or learn the mesh dynamics model [8, 9], which are not feasible in the real world. Other methods, i.e. [7, 3, 2] can perform self-supervised training in the real world. Still, their policy training relies on dense value maps which require a large amount of negative samples to achieve high generalization ability [16]. In comparison, our method simplifies the action space by firstly learning a sparse set of semantic-rich keypoint candidates from human priors and then predicting ranking scores for these

candidates. This novel design makes human-in-the-loop learning in real-world sample-efficient and scalable.

**Cloth Folding.** There have been two lines of works for cloth folding: (1) Heuristic-based methods [17, 18, 19, 20, 21, 22, 2] rely on heuristic rules to fold cloth. These methods have limited generalization ability because they usually have strong assumptions about cloth types, textures, and shapes. (2) Goal-conditioned learning-based methods [23, 24, 25, 26, 27] can perform the folding task with a pre-defined goal state, but such goal state is unavailable in the real world for a novel instance. Unlike previous works [19, 2, 1], our work integrates *unfolding* and *folding* into a unified end-to-end policy model, the **UFONet**, which can handle corner cases by continuously adding training data.

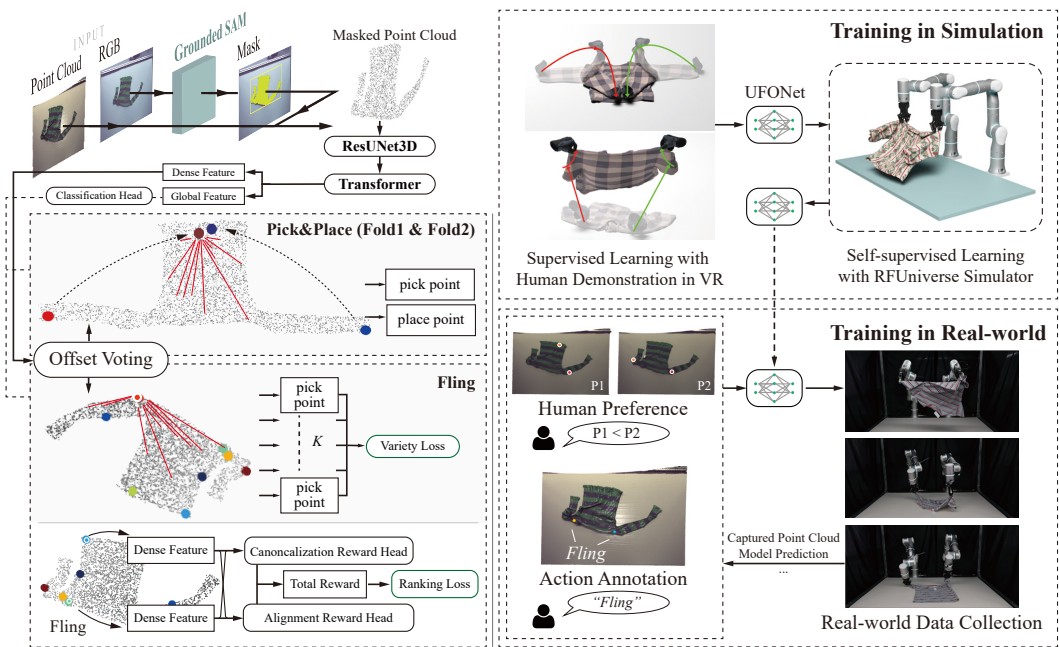

Figure 2: **Left**: UFONet takes a masked point cloud of the observed garment state as the input, predicts the primitive action type, and regresses the actioning points. **Right**: The offline and online training strategies for UFONet.

## 3 Method

Starting with an RGB-D observation $\boldsymbol{I}_0 \in \mathbb{R}^{W \times H \times 4}$ of a garment's configuration $\boldsymbol{s}_0$, UniFolding employs a dual-arm robot to sequentially transform the garment to the desired state $\boldsymbol{s}^*$ using UFONet to determine the action type $m \in \mathcal{M}$ and its parameters from evolving RGB-D observations $\boldsymbol{I}_t$. These parameters contain pick points $\boldsymbol{p}_i = (x_i, y_i, z_i)$ and place locations $\boldsymbol{q}_i = (\hat{x}_i, \hat{y}_i, \hat{z}_i)$ where $i = 1, 2$ represents the left and right arm respectively. If the points are unreachable, UniFolding employs two primitive actions, *drag* and *mop* to ensure the process works smoothly.

The pipeline is shown in Fig. 1. The primitive actions, UFONet's design, and its training are detailed in Sec. 3.1, 3.2, and 3.3. respectively.

### 3.1 Action Primitive

**Fling** The ABB YuMi [28] robot's fling operation in SpeedFolding [2] is adapted to the Flexiv Rizon [29] robot arm by modifying force thresholds, velocity, and trajectory parameters. The fling parameters are $\boldsymbol{a}_f = (\boldsymbol{p}_1; \boldsymbol{p}_2)$. The rotation angles of end-effectors are generated by simple heuristic rules to avoid collision and increase reachability.

**Pick-and-Place (fold 1 & fold 2)** Given two pick points and two place locations, the arms first pick the pick points, move to a certain height above the place locations, and release the grasp. The parameters of *pick-and-place* is $\boldsymbol{a}_{p\&p} = (\boldsymbol{p}_1, \boldsymbol{q}_1; \boldsymbol{p}_2, \boldsymbol{q}_2)$.

**Drag & Mop** If pick points or place locations are out of dual-arm reach, we change the garment position using rule-based points and trajectory, making unfolding and folding actions feasible.

## 3.2 UFONet for Garment Unfolding and Folding

For garment observation $I_t$ at time step $t$, **UFONet** will first convert $I_t$ into a point cloud $o_t$ and randomly sample it so that $o_t \in \mathbb{R}^{N \times 3}$, where $N$ is the number of points. Then, it predicts the next action type and corresponding parameters from three primitive actions: *fling*, *fold1*, and *fold2*. As the actions do not share the same parameter space, we predict $a_{f,t}$ and $a_{p\&p,t}$ in different branches. The overall framework design is illustrated in Fig. 2.

**Image Processing** We adopt the Grounded-SAM [30, 31] model to segment the RGB image from $I_t$ with prompt "cloth", multiply the mask by the depth image, and convert the masked depth to point cloud $o_t$ based on camera intrinsic parameters.

**Feature Extraction** We adopt a ResUNet3D [32] model to extract features from $o_t$. The ResUNet3D [32] model is an efficient 3-D CNN architecture based on sparse convolution that is well-suited for extracting high-resolution features from 3D data. The extracted features are then passed to a self-attention module based on Transformer [33], which processes the features and produces two sets of outputs: global features $\mathcal{F}_g \in \mathbb{R}^{128}$ and per-point dense features $\mathcal{F}_d \in \mathbb{R}^{N \times 128}$.

**Action Classification** $\mathcal{F}_g$ generated by the Transformer model is fed into a classification head which will predict a smoothed score. When the smoothed score reaches a certain threshold, the system will go into the folding stage and execute two continuous pick-and-place actions (*fold1* and *fold2*).

**Pick-And-Place Action Prediction** We predict $a_{p\&p,t}$ based on $\mathcal{F}_d$. With the definition of standard folding procedures within a category (see Appendix C), both pick points $p_t$ and place locations $q_t$ tend to concentrate on a few areas. Thus, we predict two sets (*fold1* and *fold2*) of $a_{p\&p,t}$ in this branch.

**Fling Action Prediction** We predict $a_{f,t}$ based on $\mathcal{F}_d$. Unlike the *pick-and-place* operation for folding, the pick point selection for the *fling* operation in the unfolding stage is much more ambiguous because the garment is usually in an unstructured state. It seems that we can only judge whether the fling point prediction is good until it actually executes the action. That's the main reason why previous works [7, 1] adopt the self-exploration approaches for fling point prediction. However, after analyzing the statistics from human demonstration data through VR, we surprisingly find that humans have strong preferences for the *fling* operation: humans tend to grasp semantic-rich areas such as the cuff, shoulder, waistline, *etc.* for *fling* action (see Appendix B for more details). Thus, we choose to directly learn keypoint prediction from human demonstration data. However, due to the ambiguous nature, the keypoint distribution for the *fling* is not as concentrated as the *pick-and-place* operation. Thus, we leverage the multi-modal distribution property and learn to predict $K$ sets of $p$ in this branch, where $P = \{p^{(j)}\}_{j=1,...,K}$, supervised by a variety (Minimum-over-N) loss [34]:

$$L_{kp}(P, p^*) = \min_{\{p^{(1)},...,p^{(K)}\} \in P} \left\{ d\left(p^*, p^{(1)}\right), d\left(p^*, p^{(2)}\right), ..., d\left(p^*, p^{(K)}\right) \right\}, \quad (1)$$

where $p^*$ is the human-preferred point, and $d(\cdot, \cdot)$ is the distance metric. Intuitively, $L_{kp}$ only supervises the predicted keypoint closest to the human-preferred point, which encourages the variety of the $K$ predicted keypoints.

The prediction of keypoint candidates $P$ is constructed by directly regressing 3D keypoints through attention-based offset voting [35]:

$$p^{(j)} = \frac{1}{N} \sum_{k=1}^{N} w_{k,j} \left(x_k + u_{k,j}\right), \quad s.t. \sum_{k=1}^{N} w_{k,j} = 1, \quad (2)$$

where $p^{(j)}$ is the $j$-th keypoint prediction, $w_{k,j} \in [0, 1]$ is the attention score, $x_k \in o_t$ is the $k$-th point in the input point cloud $o_t$, and $u_{k,j}$ is the 3D offsets of the $j$-th keypoint $p^{(j)}$ with respective to the $k$-th point $x_k$. The attention score $w_{k,j}$ and offsets $u_{k,j}$ are predicted by MLP with dense features $\mathcal{F}_d$ as input. Finally, we should select a keypoint pair from $P$ to obtain $a_{f,t}$. We design an

evaluation module to score any two input keypoints. Specifically, for any two points with the indices of $j$ and $k$ in $\boldsymbol{P}$, we generate embeddings by Eq. 3:

$$\boldsymbol{e}_{j,k} = \text{MLP}([\boldsymbol{F}_j, \boldsymbol{p}^{(j)}, \boldsymbol{F}_k, \boldsymbol{p}^{(k)}]), \tag{3}$$

where $\boldsymbol{F}$ is the feature vector, defined as the weighted sum from the per-point dense feature $\mathcal{F}_d$. In practice, we find that regressing keypoint candidates in canonical space [36] is much easier than regressing them directly in task space. Please see Appendix J for the detailed version of the *fling* action prediction formulation.

Inspired by ClothFunnels [1], we predict two factorized Q-value scores given the embedding $\boldsymbol{e}_{j,k}$ as input, namely **Canoncalization score** $R_\text{C}$ and **Alignment score** $R_\text{A}$. Please see Appendix D for more details on $R_\text{C}$ and $R_\text{A}$. Finally, we calculate the total score $R_\text{CA}$ by Eq. 4:

$$R_\text{CA} = (1 - \beta)R_\text{C} + \beta R_\text{A}. \tag{4}$$

Here $\beta$ is a balance factor that can be further optimized during the real-world fine-tuning process. In the inference phase, we calculate $R_\text{CA}(\boldsymbol{e}_{j,k})$ for all $K(K-1)/2$ pairs of keypoint candidates, and choose the pair with the highest $R_\text{CA}$ score as the final pick points.

### 3.2.1 Discussion: Action Poses Beyond Reachability

Previous methods [2, 1, 7] often use reachability masks to filter out unreachable poses on the action value map. This approach is effective for small or specific garments, but it could filter out optimal action predictions for garments of varied shapes and large sizes. To resolve this problem, our model utilizes an active movement strategy: when the policy model's optimal action points are unreachable, it automatically switches to *drag* (in *unfolding* stage) or *mop* (in *folding* stage). *drag*'s pick points are selected from the lines that connect the optimal points to the robot bases, while *mop*'s are chosen via simple heuristics, as folded garments are typically well-shaped. These actions reposition the garment until it's a fixed distance away from the dual-arm robot.

### 3.3 Data Collection & Network Training

**Supervised Training with Human Demonstrations in VR** We first train UFONet with the human demonstration data in VR. The keypoint candidates for *fling* action are supervised with variety loss defined in Eq. 1. The grasping and releasing points of *pick-and-place* action are supervised with Smooth-L1 loss. The VR dataset of human demonstrations contains 1218 manipulation videos for 203 short-sleeve T-shirts and 1575 videos for 315 long-sleeve shirts. Each video contains $\sim 5$ action steps on average to fully smooth and fold the garment. The total data collection time in VR takes about 16 hours. The training time for supervised training takes about 4 hours on one single RTX 3090 GPU for each category. Please see Appendix G for more details of the VR data collection.

**Self-supervised Training for Unfolding** As shown in Fig. 2 (right), we continue to perform self-supervised training in simulation on the pre-trained model from supervised learning. It is used for training the score prediction head for *fling* action in Fig. 2 (left). The score $R_\text{CA}$ is supervised with Smooth-L1 loss. We use RFUniverse [37] as the simulation environment, and the cloth simulation is based on ClothDynamics [12], a GPU-based cloth-specific physics engine in Unity [38]. The garment mesh models for simulation are selected from the CLOTH3D [39] dataset. The training process in this stage is similar to ClothFunnels [1], except that our training is based on a pretrained model and we only need to choose pick points from a sparse set of keypoint candidates. Thus, our training is surprisingly sample-efficient, which only takes about 12 hours for data collection and model training on one single RTX 4090 GPU. In comparison, the training process of ClothFunnels [1] takes 2 days with 4 RTX 3090 GPUs. Please refer to Appendix H for more details.

**Real-world Fine-tuning** We develop an online learning framework for real-world fine-tuning. As shown in Fig. 2, we first use the current policy model in each episode to automatically collect data for garment *unfolding* and *folding*. The human annotators will simultaneously annotate the collected data samples, which will be used for training the policy model for the next episode. Specifically, the human annotators provide their preferences on (1) the best action type and actioning points for

the current garment state. (2) comparison of the keypoint candidates (for *fling* action only). As the ground-truth $R_{\text{CA}}$ score cannot be obtained in real-world scenarios, we fine-tune the score model with human-in-the-loop learning. Given $T$ randomly selected keypoint pairs from all possible pairs, the annotators rank the pairs by making $M$ comparisons. For each comparison, we denote the keypoint pair $\sigma$, and the two pairs to be compared $\sigma_1$ and $\sigma_2$. The annotator gives a label $\mu$, where $\mu \in \{(0,1),(1,0),(0.5,0.5)\}$. The odds that $\sigma_1$ is superior than $\sigma_2$ is calculated by Eq. 5:

$$\hat{P}\left[\sigma_1 \succ \sigma_2\right] = \exp(\hat{R}_{\text{CA}}(\sigma_1))/(\exp(\hat{R}_{\text{CA}}(\sigma_1)) + \exp(\hat{R}_{\text{CA}}(\sigma_2))). \tag{5}$$

We use the following cross-entropy loss [40] in Eq. 6 to supervise the evaluation module, with the collection of annotations denoted as $A$:

$$\text{loss}(\hat{r}) = - \sum_{(\sigma_1,\sigma_2,\mu)\in A} \mu(1)\log \hat{P}\left[\sigma_1 \succ \sigma_2\right] + \mu(2)\log \hat{P}\left[\sigma_2 \succ \sigma_1\right]. \tag{6}$$

In total, we collected 2432 data samples in the real world for the two categories and annotated $M = 16$ comparisons for each data sample. The whole data collection and annotation process takes about 20 hours. Please see Appendix I for more details on the process of human preference annotation and learning.

## 4 Experiment Setup

### 4.1 Garments in Real World and Simulation

We examined two categories of clothing: long-sleeve shirts and short-sleeve T-shirts, incorporating 60 diverse real-world samples. These varied in size ($38cm \times 60cm$ to $80cm \times 167cm$), aspect ratios ($0.2695 : 1$ and $1.1167 : 1$), and materials (cotton, polyester, spandex, nylon, viscose, wool, etc.). See Appendix K for more details. The garments were divided into train/test sets at $2 : 1$ ratio, with real-world fine-tuning garments selected from the training set, and the testing set garments remaining unseen in all experiments. Fig. 4 (b) showcases all the garments in the test set. For simulation experiments, we split the instances in CLOTH3D [39] into train/test sets at $9 : 1$ ratio.

### 4.2 Robots Setup

Fig. 4 shows our setup: two Flexiv Rizon [29] robots with AG-95 grippers [41], equipped with force sensors for stretching cloth, are placed at a rigid Polypropylene (PP) board table. A high-precision depth camera, Photoneo MotionCam3D M+ [42], and an RGB camera, MindVision SUA202GC, are mounted above. The Flexiv control API's *Contact Grasp* function allows for adaptive control of gripper height during operations, making it ideal for use on hard tables. We randomly generate crumpled garment states for data collection and testing by grasping a random point and lifting at random heights ranging from 0.5m to 1.0m. A grasp failure detection mechanism is implemented to perform automatic re-grasping and lifting action if required.

## 5 Experiment Results

### 5.1 Metrics

**IoU (Intersection over Union)** IoU measures the garment unfolding quality by comparing the mask with the target T-shaped mask. This metric is used to evaluate the unfolding quality.

**Normalized Coverage** This is the ratio of the current top-down pixel count of the garment mask to the maximum count at the target T-shaped pose, which can be used to evaluate the unfolding quality.

**Success Rate for End-to-end Unfolding and Folding** Success rate averages over 10 trials for each garment. Each experimental trial begins with a randomly crumpled garment, and success is determined by first smoothing out the garment, and then folding it according to predefined rules within 10 action steps. Refer to Appendix C for additional details on the folding rules.

### 5.2 Unfolding and Folding Results

**Comparison with baselines.** Tab. 1 and Fig. 3 shows the quantitative and qualitative results for the unfolding and folding tasks. We use the pre-trained model from ClothFunnels [1] as the

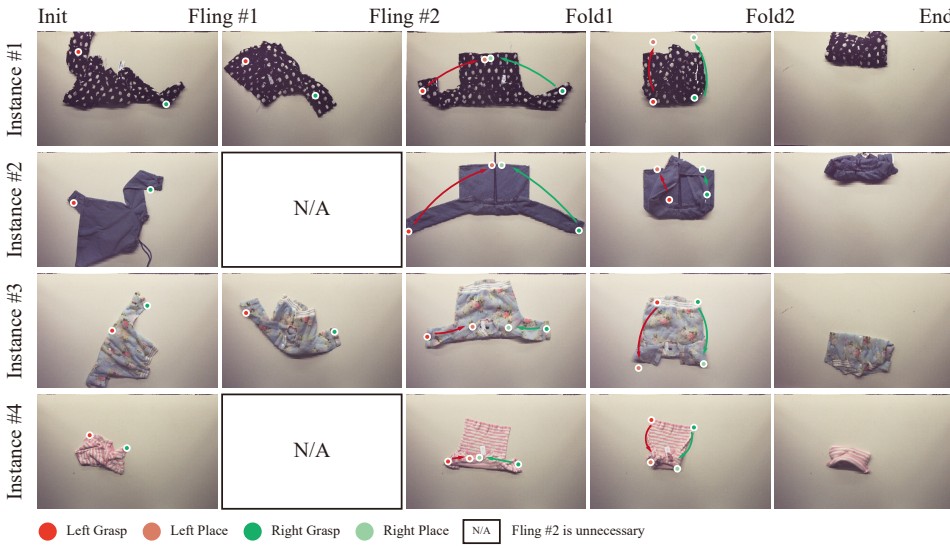

Figure 3: The figure illustrates the shape transformations of different types and sizes of clothes after applying each primitive action of the UniFolding system under various initial states.

Table 1: The system-level comparison of UniFolding and ClothFunnels [1] in the unfolding and folding process for each **unseen** garment in the testing set.

| Metric | Method | Garment ID (Long-sleeve Shirts) | | | | | | | | | | |
|--------|--------|------|------|------|------|------|------|------|------|------|------|------|
| | | #1 | #2 | #3 | #4 | #5 | #6 | #7 | #8 | #9 | #10 | Mean |
| **IoU** | Ours | 0.572 | **0.384** | 0.393 | **0.639** | **0.556** | **0.411** | 0.440 | **0.454** | 0.408 | **0.649** | **0.491±0.098** |
| | ClothFunnels [1] | **0.601** | 0.356 | **0.463** | 0.384 | 0.339 | 0.393 | **0.505** | 0.386 | **0.419** | 0.427 | 0.427±0.074 |
| **Coverage** | Ours | 0.651 | **0.607** | **0.669** | **0.751** | **0.682** | **0.652** | 0.651 | **0.586** | **0.641** | **0.714** | **0.660±0.045** |
| | ClothFunnels [1] | **0.658** | 0.591 | 0.581 | 0.385 | 0.389 | 0.453 | **0.679** | 0.350 | 0.623 | 0.526 | 0.524±0.115 |
| **Success** | Ours | **7/10** | **6/10** | **8/10** | **10/10** | **6/10** | **6/10** | **8/10** | **6/10** | **8/10** | **8/10** | **73±13%** |
| | ClothFunnels [1] | **7/10** | 3/10 | 3/10 | 0/10 | 2/10 | 2/10 | 0/10 | 0/10 | 4/10 | 3/10 | 24±21% |

| Metric | Method | Garment ID (Short-sleeve T-shirts) | | | | | | | | | | |
|--------|--------|------|------|------|------|------|------|------|------|------|------|------|
| | | #11 | #12 | #13 | #14 | #15 | #16 | #17 | #18 | #19 | #20 | Mean |
| **IoU** | Ours | 0.658 | 0.674 | 0.675 | 0.735 | 0.601 | 0.595 | 0.691 | 0.743 | 0.700 | 0.637 | 0.670±0.047 |
| **Coverage** | Ours | 0.737 | 0.709 | 0.718 | 0.768 | 0.637 | 0.658 | 0.686 | 0.734 | 0.697 | 0.672 | 0.701±0.031 |
| **Success** | Ours | 5/10 | 4/10 | 6/10 | 8/10 | 6/10 | 2/10 | 8/10 | 8/10 | 7/10 | 6/10 | 60±18% |

baseline for comparison (it only has the model for long-sleeve shirts). We can see from Tab. 1 that ClothFunnels [1] has generalization problems on our challenging test garments. It works relatively well for garments with solid and light color (*e.g.*, garment #1, #9 in Fig. 4), but suffers from complex textures (*e.g.*, garment #4 in Fig. 4), dark colors (*e.g.*, garment #5, #8) and unusual shapes (*e.g.*, garment #2 with spindly sleeves). Specifically, most failure cases of ClothFunnels [1] come from two sources: (1) the abnormal *fling* action prediction in the unfolding process (*e.g.*, grasping two points on one single sleeve) (2) non-ideal keypoint prediction in the heuristic folding process for garments with complex textures (*e.g.*, garment #7 has high IoU and Coverage but low folding success rate). In comparison, our method has better mean performance and robustness both on

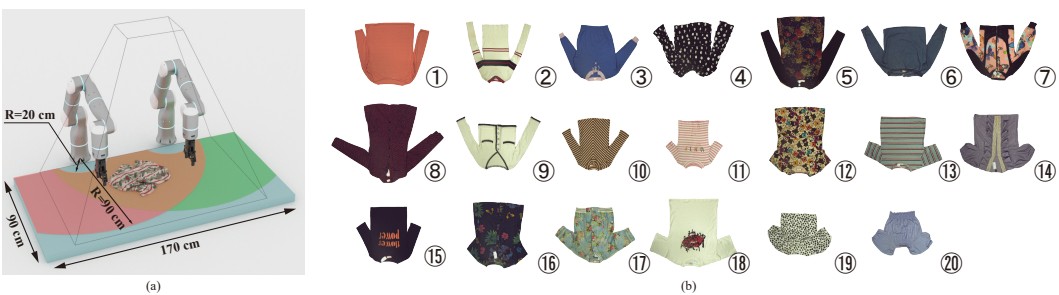

Figure 4: Real-world hardware setup and **unseen** garment instances for testing. The field of view boundaries of camera system are marked by black indicators.

metrics of unfolding (*e.g.*, IoU, Coverage) and overall success rates (see Tab. 1). We have also compared the heuristic folding policies (*i.e.*, keypoint detector in ClothFunnels [1] and template matching in SpeedFolding [2]) with our learned folding policy. Please see Fig. 6 and Fig. 7 in Appendix A for more corner cases that such heuristic folding policies can not handle.

**Discussion for different garments.** It is worth noting that for some garments (#3, #7, #9) with very long sleeves, our method has lower IoU but a higher success rate because our fine-tuned policy model tends to grasp two cuffs directly. Our dual-arm robots cannot extend long enough to stretch the garment fully (see row 2, column 3 in Fig. 3 for an example). Fortunately, such a state does not influence much the subsequent folding process. Besides, the success rate of folding short-sleeved T-shirts is considerably lower than that of long-sleeved shirts, despite having higher IoU and Coverage. This is due to the high flatness requirements in the folding process of short sleeves. If the sleeves are curled up or covered, they may not affect the IoU or Coverage significantly, but they can cause significant difficulties in the subsequent folding process. This effect is noticeable for T-shirts with extremely short sleeves (e.g. garment #12, #16). Thus, better metrics for evaluating unfolding performance are desired. We also observe a large variance in success rate on garments within and across categories, which indicates that the shape and physical material could greatly affect the difficulty of subsequent folding.

## 5.3 Sample Efficiency and Scalability

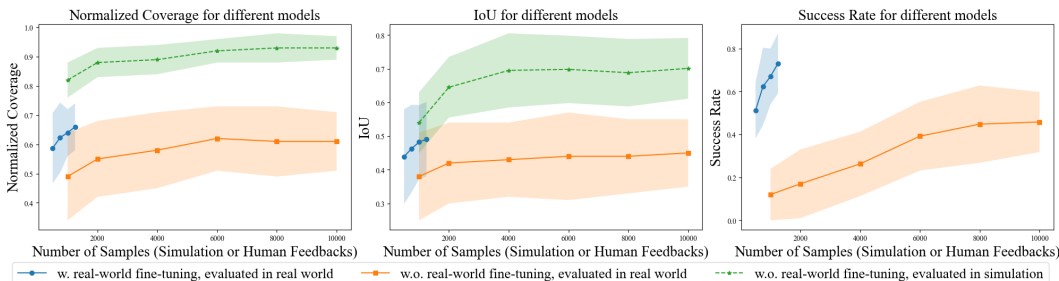

Figure 5: Variation in model performance with varying training sample sizes in both simulation and real-world settings, evaluated using long-sleeve shirts. The number of samples indicates the volume of data in self-supervised learning for simulation or human feedback for real-world fine-tuning.

Fig. 5 shows the performance of models trained with different numbers of samples. We can see that the models only trained in simulation (without real-world fine-tuning) have very large sim2real gaps on our test garments. We believe that both our method and ClothFunnels [1] suffer from the inaccurate dynamics of real-time cloth simulators [10, 12]. However, with a very limited number ($\sim$ 1200) of real-world fine-tuning data samples, the model performance in real world increases rapidly, which proves the sample efficiency of our human-in-the-loop fine-tuning process.

### 5.4 Limitations and Failure Cases

In our current implementation, if the grasping point on the garment has multiple layers, the robot gripper can NOT only grasp a single layer of cloth. In summary, four common failure cases relate to this problem: (1) Self-entanglement state. (2) The garment is folded in half. (3) The front and back of the garment are separated. (4) Garments with open zippers or buttons. Please see Fig. 11(a) in the appendix for more visualizations of failure cases.

## 6 Conclusion

In this work, we propose a novel system, UniFolding, to address the significant challenges associated with automating garment unfolding and folding. This system leverages an end-to-end neural network, UFONet. Our system is data-efficient, thanks to our human-centric data collection and training pipeline. It is scalable, owing to the unified policy model and data-driven paradigm, and it is generalizable, given its ability to handle garments with large variations in sizes, shapes, textures, and materials. We believe UniFolding is on track toward achieving full automation of the robotic garment folding task. In the future, we are interested in extending the capabilities of the UniFolding system to accommodate more garment categories.

**Acknowledgments**

We thank Wei Jiang for verifying the baseline model and paper writing. We thank Yibo Shen, Jieyi Zhang, Tutian Tang, and Wenxin Du for helpful discussions on cloth simulators. This work was supported by the National Key R&D Program of China (No. 2021ZD0110704), Shanghai Municipal Science and Technology Major Project (2021SHZDZX0102), Shanghai Qi Zhi Institute, and Shanghai Science and Technology Commission (21511101200).

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

# Appendix A    Comparison between Learned Folding Policy and Heuristic Folding Policy

In this study, we undertook comparative evaluations of ClothFunnels [1], SpeedFolding [2], and UFONet, centering on the strengths and weaknesses of their respective folding policies.

**Comparison with ClothFunnels [1].** We visually depicted their inference outcomes for long-sleeves, with specific emphasis on ClothFunnels' [1] keypoint detection results and UFONet's predicted grasping points. For a comprehensive understanding of the distinct features of both methodologies, a selection of eight garments was employed as illustrative examples in Fig. 6.

We found that ClothFunnels [1] exhibits a relatively good performance when dealing with solid and light colored garments, but it tends to make erroneous predictions for garments with complex textures. Such predictions often lead to overlapping or missing key points. Conversely, UFONet is capable of making reasonable predictions for a broader range of garments.

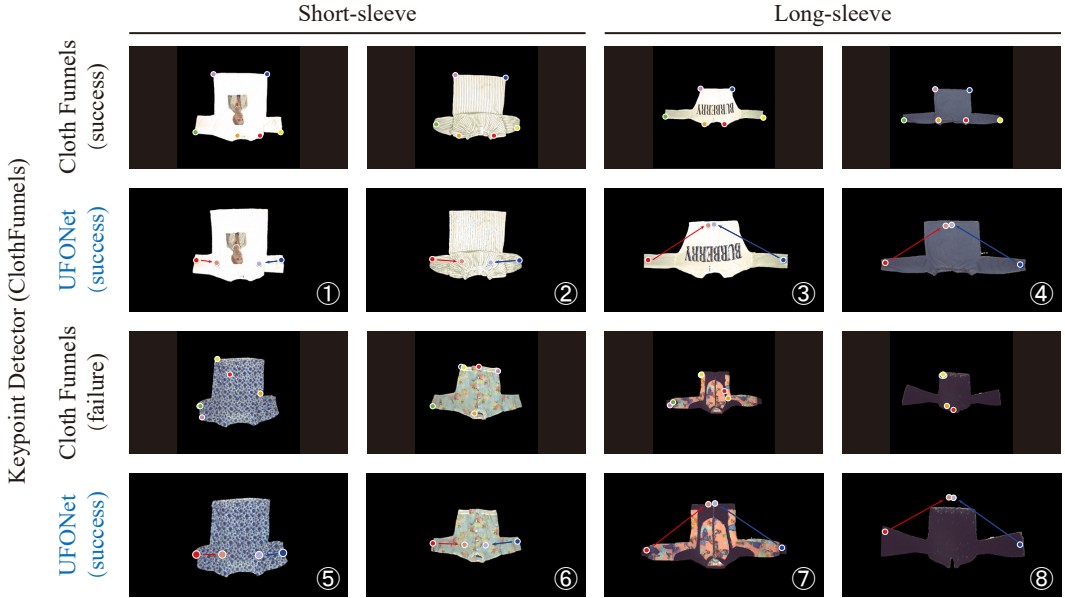

Figure 6: This figure illustrates how the folding policy in ClothFunnels [1] and UFONet behave differently in 8 cases, including 4 short-sleeves and 4 long-sleeves. The cases numbered from 1-4 are where both ClothFunnels [1] and UFONet give correct results. The cases numbered from 5-8 are where the standalone keypoint detector in ClothFunnels [1] failed to predict correct keypoints for heuristic folding but UFONet outputs correct grasp points and place points. The keypoint colors in the visualization figure for ClothFunnels [1] indicates the keypoint index (r.g. left cuff or right cuff). The wrong prediction of these keypoints could make the following heuristic folding fail.

**Comparison with SpeedFolding [2].** We visually depicted their inference outcomes for short-sleeves, with specific emphasis on SpeedFolding's [2] template matching results and UFONet's predicted grasping points. For a comprehensive understanding of the distinct features of both methodologies, another selection of eight garments was employed as illustrative examples in Fig. 7.

We have found that SpeedFolding is able to provide relatively accurate predictions and generate the correct folding lines for regular garments that conform to its templates. However, for irregularly shaped garments, the predictions given by SpeedFolding often have incorrect rotations and translations. In contrast, UFONet is able to handle these irregularly shaped garments more effectively.

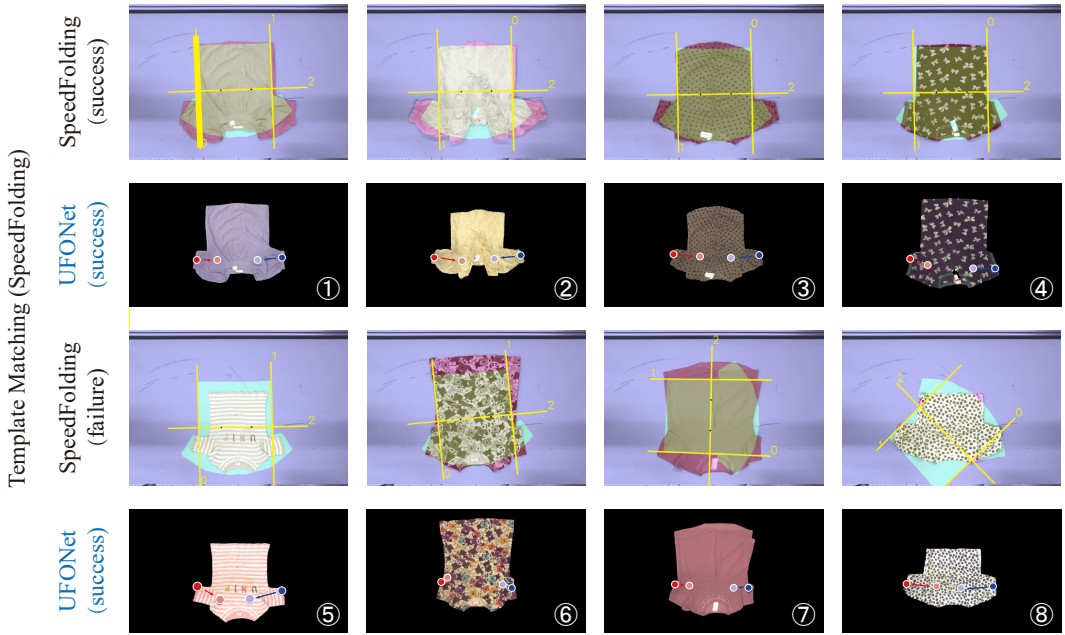

Figure 7: This figure illustrates how the folding policy in SpeedFolding [2] and UFONet behave differently in 8 cases (short-sleeves only). The cases numbered from 1-4 are where both SpeedFolding [2] and UFONet gives correct results. The cases numbered from 5-8 are where SpeedFolding failed to match the template correctly but UFONet outputs correct grasp points and place points.

## Appendix B    Evidence of Human Preferences in *fling* Action

Fig. 8 and Fig. 9 show grasping point distribution (showed in NOCS [36] space) of human demonstration data collected by VR. We can see that humans frequently grasp shoulders, collars, and waists in the earlier stage of the unfolding process when the garment is usually more crumpled. Humans will probably grasp shoulders at the later stage of the unfolding process when the garment is more flattened and recognizable.

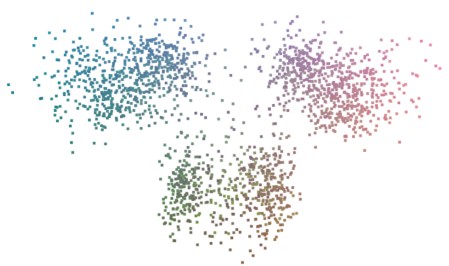

Figure 8: The grasping point distribution (showed in NOCS [36] space) for *fling* action in human demonstration data through VR. These points are from **earlier** steps of the unfolding process.

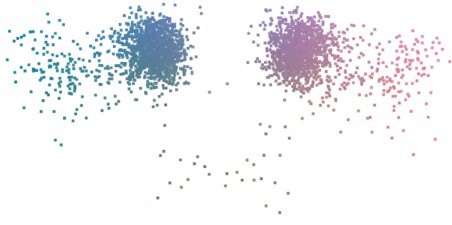

Figure 9: The grasping point distribution (showed in NOCS [36] space) for *fling* action in human demonstration data through VR. These points are from **later** steps of the unfolding process.

## Appendix C   Folding Rules

Fig. 10 shows the general folding rules to generate sub-goal targets which will be shown to the human demonstrators and evaluators.

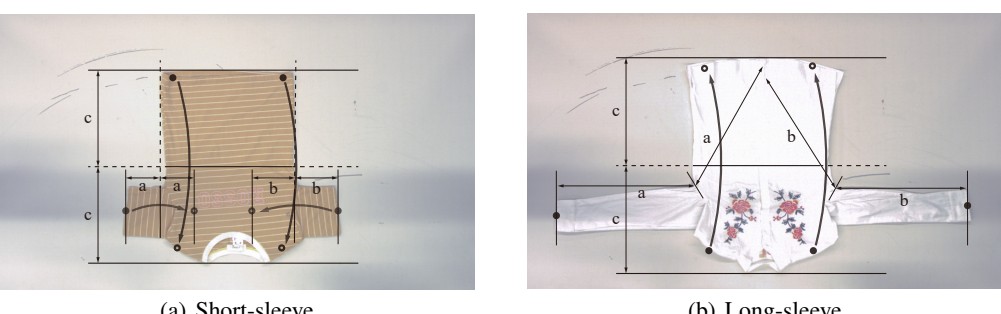

(a) Short-sleeve                                   (b) Long-sleeve

Figure 10: This figure shows how short-sleeves and long-sleeves are folded to generate sub-goal targets which will be shown to the human demonstrators and evaluators. In the first folding step, we will fold the two sleeves (a, and b in the figure) simultaneously according to the folding line. In the second folding step, we will fold the garment in half (c in the figure) according to the folding line. It is worth noting that this figure is only an illustration or guidance that lets the volunteers get a sense of how the garment could be folded, the exact locations of the grasping points and placing points are still generated by humans through VR.

## Appendix D   How $R_{\mathrm{C}}$ and $R_{\mathrm{A}}$ are Calculated

Intuitively speaking, $R_{\mathrm{C}}$ encourages actions that make the garment more flattened and more similar to the canonical pose, and $R_{\mathrm{A}}$ encourages actions that make the garment more aligned with the target pose in planar position and rotation. Please refer to ClothFunnels [1] for the detailed definition of $R_{\mathrm{C}}$ and $R_{\mathrm{A}}$.

## Appendix E   Limitations and Failure Cases

In our current implementation, if the grasping point on the garment has multiple layers, the robot gripper can NOT only grasp a single layer of cloth. **1. Self-entanglement state**. In practice, we find that only relying on *Fling* action can not fully flatten the garment in a self-entanglement state. More dexterous manipulation skills are required for this problem. **2. The garment is folded in half**. Most attempts to manipulate garments in this folded state will always grasp two layers of cloth, which may be stuck in a loop forever and fail to finish the unfolding task. **3. The front and back of the garment are separated**. In this state, the robot should accurately grasp the recognizable cuff with only one single layer of cloth to fully smooth the garment. However, such behavior is hard to

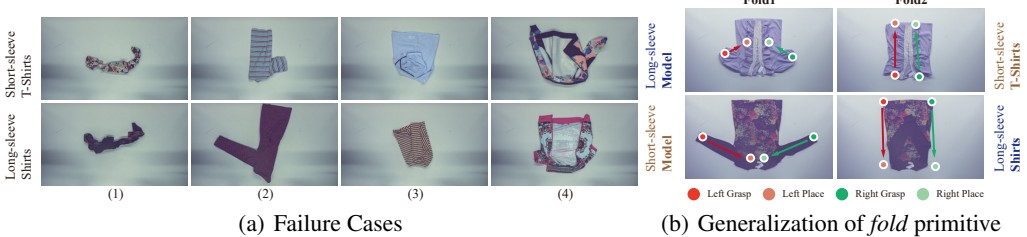

(a) Failure Cases                 (b) Generalization of *fold* primitive

Figure 11: (a) Failure cases and (b) *fold* primitive generalization across categories.

accomplish in our system. **4. Garments with open zipper or buttons**. In order to fully smooth garments with open zippers or buttons, more dexterous manipulation skills are required. Please see Fig. 11(a) in the appendix for more visualizations of failure cases.

## Appendix F    Generalization between categories

Flingbot [7] has proved that *fling* action could be transferred between objects with different shapes (*e.g.*, tower and T-shirt). What about transferring the learned folding actions between categories? In our setting, the original folding actions are slightly different for short-sleeve and long-sleeve shirts (*e.g.*, the folding direction of *fold2* are opposite to each other). We swap the pre-trained UFONet models for long-sleeved shirts and short-sleeved T-shirts, respectively attempting to predict the action poses of *fold1* and *fold2* in the other category. The visualization results of model predictions are shown in Fig. 11(b). We can see that the learned *fold* primitives can be directly transferred to other categories with different shapes, and they can even create new folding patterns in this way (*e.g.*, the long sleeves are folded towards the collar rather than the waist).

## Appendix G    Details of Human Demonstration Data Collection in VR

**VR Recording System**    We build a real-time data recording system for collecting human demonstration data for garment manipulation in Virtual Reality. This system is based on the VR-Garment system implemented in GarmentTracking [13]. It is driven by Unity, and the physics engine for cloth simulation is based on Obi [11]. In practice, this system can effectively collect large amounts of human demonstration data for thousands of garments with different shapes and sizes.

**Data Recording Pipeline**    The data recording pipeline is similar to that in GarmentTracking [13]. Firstly, the volunteer will put on an HTC Vive Pro VR Headset and VRTRIX VR gloves. Secondly, a virtual garment from the CLOTH3D [39] dataset will randomly drop on the table in virtual space. Thirdly, the volunteer will use his hands to perform the action primitives defined in the main paper for multiple steps to fully smooth and fold the garment. On average, the whole multi-step manipulation process for one garment only takes about 20s in VR.

**Data Post-processing**    The raw data generated by the data recording pipeline are videos that contain the garment mesh vertices and hand poses of each frame. We use a simple method to automatically convert hand poses into robot gripper poses. After data recording, We will perform the following data post-processing steps to generate data that are available for network training: Firstly, we automatically divide the whole video of the garment manipulation process into multiple valid action intervals. The start and ending of each action interval are decided by the grasping and releasing states of both human hands. Secondly, we use simple rules to automatically generate labels of action primitive type for all valid action intervals based on patterns of human actions. Thirdly, we re-render the garment mesh in Unity and generate RGB-D image, mask, NOCS [36] map, and gripper poses for the starting frame of each action interval.

## Appendix H   Details of Self-supervised Learning in Simulation

The initial garment state for each experiment trial in simulation is generated by two ways: (1) *random lift*: randomly grasp one point on the garment and lift it in the air to generate crumpled state. (2) *random pick-and-place*: randomly perform one pick-and-place action for one random corner (*e.g.*, cuff, waist) on one fully smoothed garment. In the data collection process, we randomly choose from these two ways to generate initial garment state with probability $30\%$ for *random pick-and-place* and $70\%$ for *random lift*. For each action step, the model will randomly explore and select one random pair of keypoint candidates for *fling* action with probability $p = 80\%$, otherwise it will execute the best action prediction.

The training in this stage rely on an initial model with folding action prediction branches from supervised-training with VR data, and folding data is not available in this stage. Thus, how to avoid the model from forgetting is a non-trivial problem. We devise a novel hybrid-training strategy for self-supervised training, which allows us to unlock all the model parameters during training without hurting the folding performance. Specifically, we mix the data samples of human demonstrations in VR with the data samples collected via self-exploration in simulation. During the training process, we will perform two separate forwarding processes for two different data sources (Demonstrations and Exploration Memory) and then calculate the losses for these two data sources separately. In the backward process, the two losses will be added together and the gradients will be back-propagated and accumulated on the same model weights. We use a relatively small learning rate for the training of demonstration data, so the performance on folding could remain the same level during the training process. We use PytorchLightning to implement the hybrid training strategy.

For initial state of *random pick-and-place*, we generate additional data of best grasping points for *fling* action with simple heuristic rules because such state is usually well-shaped. These data samples will be used to aid the training of keypoint candidate prediction branch under simple and structured garment states.

## Appendix I   Details of Human Preference Annotation and Learning

During the data collection process, for each action step, the model will randomly select a pair of keypoint candidates for *fling* action with $p = 5\%$ probability, otherwise it will select the best action prediction and execute the action. We have developed an online data annotation system which allow multiple users to annotate the newest data samples generated from the robots and save them into the database. The annotation process and the robot data collection proceed simultaneously in the real world. In practice, we annotate 16 comparisons from the top $20\%$ keypoint candidates ranked by $R_{CA}$ scores for each data sample, which can filter out most of the bad keypoint candidate combinations. Besides, the system will additionally generate the comparisons between human-annotated best action points and all other keypoint candidates. In practice, it is slightly faster for one human annotator to annotate one data sample than executing one action step with robots. In fact, the main bottleneck of the data collection process is the action execution speed of real robots rather than human annotators.

The training of the online learning stage adopts a hybrid-training strategy similar to that in Appendix H which takes both the self-supervised data in simulation and human feedbacks in the real world as input. The losses will be calculated for these two branches separately and the gradients will be accumulated together to perform the parameter updating. In the online learning stage, the balance factor $\beta$ in Eq. 4 for each pair of keypoint candidatates is predicted by a MLP branch.

## Appendix J   Details of Keypoint Prediction for *fling* action

The dense features generated by the Transformer model will be used for the pose prediction branch for *fling* action. This branch will predict two grasp points for *fling* action. The grasp point indicates the location on the garment where the robot should grip and perform the flinging action.

**Keypoint Candidate Prediction**   Humans will frequently grasp recognizable keypoints on the garment (e.g. cuff, shoulder, waist) for *fling* action. Motivated by this observation, we choose to directly learn possible keypoint candidates purely from human demonstration data. However, the distribution of these keypoint candidates on the garment is multi-modal, so we firstly predict $K$ possible keypoint candidates $\boldsymbol{P} = \{\boldsymbol{p}^{(j)}\}_{j=1,...,K}$ , then supervise them with the variety (Minimum-over-N) loss [34] in Eq. 7:

$$L_{kp}(\boldsymbol{P}, \boldsymbol{p}^*) = \min_{\{\boldsymbol{p}^{(1)},...,\boldsymbol{p}^{(K)},\} \in \boldsymbol{P}} \left\{ d\left(\boldsymbol{p}^*, \boldsymbol{p}^{(1)}\right), d\left(\boldsymbol{p}^*, \boldsymbol{p}^{(2)}\right), \ldots, d\left(\boldsymbol{p}^*, \boldsymbol{p}^{(K)}\right) \right\} \tag{7}$$

where $\boldsymbol{p}^*$ is the human-preferred point, and $d(\cdot, \cdot)$ is the distance metric. Intuitively, $L_{kp}$ only supervises the predicted keypoint closet to the ground-truth keypoint, which encourages the variety of the $K$ predicted keypoints. For *fling* action, we have two ground-truth keypoints $\{\boldsymbol{p}^*_{left}, \boldsymbol{p}^*_{right}\}$ for dual-arm robots, so the final loss is shown in Eq. 8:

$$L_{kp}(\boldsymbol{P}, \boldsymbol{p}^*_{left}, \boldsymbol{p}^*_{right}) = (L_{kp}(\mathcal{P}, \boldsymbol{p}^*_{left}) + L_{kp}(\mathcal{P}, \boldsymbol{p}^*_{right}))/2 \tag{8}$$

As for the prediction of keypoint candidates $\boldsymbol{P}$, an intuitive way is to use attention-based offset voting [35] to directly regress keypoints in 3D task space (the coordinate frame of the input point cloud) as shown in Eq. 9:

$$\boldsymbol{p}^{(j)} = \frac{1}{N} \sum_{k=1}^{N} \omega_{k,j} \left(\boldsymbol{x}_k + \boldsymbol{u}_{k,j}\right), \quad s.t. \sum_{k=1}^{N} \omega_{k,j} = 1 \tag{9}$$

where $\boldsymbol{p}^{(j)}$ is the $j$-th keypoint prediction, $\omega_{k,j} \in [0, 1]$ is the attention score, $\boldsymbol{x}_k \in \boldsymbol{o}_t$ is the $k$-th point in the input point cloud $\boldsymbol{o}_t$, and $\boldsymbol{u}_{k,j}$ is the 3D offsets of the $j$-th keypoint $\boldsymbol{p}^{(j)}$ respective to the $k$-th point $\boldsymbol{x}_k$. The attention score $\omega_{k,j}$ and offsets $\boldsymbol{u}_{k,j}$ are predicted by MLP with dense features generated by Transformer $\mathcal{F}_d$ as input. Finally, we should select a keypoint pair from $\boldsymbol{P}$ to obtain $\boldsymbol{a}_{f,t}$. We design an evaluation module to score any two input keypoints. Specifically, for any two points with the indices of $j$ and $k$ in $\boldsymbol{P}$, we generate embeddings by Eq. 10:

$$\boldsymbol{e}_{j,k} = \text{MLP}([\boldsymbol{F}_j, \boldsymbol{p}^{(j)}, \boldsymbol{F}_k, \boldsymbol{p}^{(k)}]), \tag{10}$$

where $\boldsymbol{F}$ is the feature vector, defined as the weighted sum from the per-point dense feature $\mathcal{F}_d$. In practice, we find that regressing keypoint candidates in canonical space [36] is much easier than regressing them directly in task space.

**Prediction in Canonical Space**   In practice, we find that regressing keypoint candidates in canonical space (Normalized Object Coordinate Space, NOCS [36]) is much easier than regressing them directly in task space. So we additionally predict per-point NOCS coordinate $\boldsymbol{c}_k \in \mathcal{C}$ for the input point cloud with dense features generated by the Transformer. Due to the bilateral symmetry property of most garments, we use the symmetric Huber loss defined in Eq. 11 to supervise NOCS prediction $\boldsymbol{C}$:

$$L_{nocs}(\boldsymbol{C}, \mathcal{C}^*) = \min\{\frac{1}{N} \sum_{k=1}^{N} Huber(\boldsymbol{c}_k, \boldsymbol{c}_k^*), \frac{1}{N} \sum_{k=1}^{N} Huber(\boldsymbol{c}_k, \boldsymbol{c}_k^{*sym})\} \tag{11}$$

where $\boldsymbol{c}_k^* \in \mathcal{C}^*$ is the original ground-truth NOCS coordinate of $k$-th point, and $\boldsymbol{c}_k^{*sym}$ is the symmetrical ground-truth NOCS target of $k$-th point.

Then we can modify Eq. 9 by replacing $\boldsymbol{x}_k$ with $\boldsymbol{c}_k$ to generate $K$ keypoint predictions $\boldsymbol{p}_{nocs}$ in canonical space instead of task space, which is shown in Eq. 12:

$$\boldsymbol{p}_{nocs}^{(j)} = \frac{1}{N} \sum_{k=1}^{N} \omega_{k,j} \left(\boldsymbol{c}_k + \boldsymbol{u}_{k,j}\right), \quad s.t. \sum_{k=1}^{N} \omega_{k,j} = 1 \tag{12}$$

Next, we need to find the corresponding 3D location $\boldsymbol{p}^{(j)}$ in task space for $j$-th keypoint from NOCS coordinate $\boldsymbol{p}_{nocs}^{(j)}$ in canonical space. Due to the local similarity of the NOCS coordinates, we can calculate $\boldsymbol{p}^{(j)}$ by weighted sum defined in Eq. 13:

$$\boldsymbol{p}^{(j)} = \frac{\sum_{k=1}^{N} \beta_{k,j} \boldsymbol{x}_k}{\sum_{k=1}^{N} \beta_{k,j}}, \quad \beta_{k,j} = \exp\left(-\alpha \cdot \left\|\boldsymbol{p}_{nocs}^{(j)} - \boldsymbol{c}_k\right\|_2\right) \tag{13}$$

Intuitively, $\beta_{k,j}$ is the weight based on the L2-distance between $j$-th keypoint $\boldsymbol{p}_{nocs}^{(j)}$ and $k$-th point $\boldsymbol{c}_k$ in canonical space. The larger $\beta_{k,j}$ is, the more likely $j$-th keypoint $\boldsymbol{p}^{(j)}$ is closer to the $k$-th point $\boldsymbol{x}_k$ in task space. We set $\alpha = 50$ by default.

Finally, we can supervise $K$ keypoint candidate predictions both in canonical space and task space by Eq. 14:

$$L_{kp\_all}(\boldsymbol{P}_{nocs}, \boldsymbol{P}, \boldsymbol{p}_{nocs}^*, \boldsymbol{p}^*) = L_{kp}(\boldsymbol{P}_{nocs}, \boldsymbol{p}_{nocs}^*) + L_{kp}(\boldsymbol{P}, \boldsymbol{p}^*) \tag{14}$$

## Appendix K    Additional Garment Details

This section presents the parameters of the garments that are used in our experiment. We use a total of 60 garments, divided into two sets: a test set of 10 long-sleeved and 10 short-sleeved garments, and a training set of 20 long-sleeved and 20 short-sleeved garments. The garments cover various materials and textures. Each garment is assigned a unique ID, and its size and material are also listed in the table. The size information indicates the height and width of the garment when fully unfolded. In addition, we capture an RGB image of each garment from a top-down view.

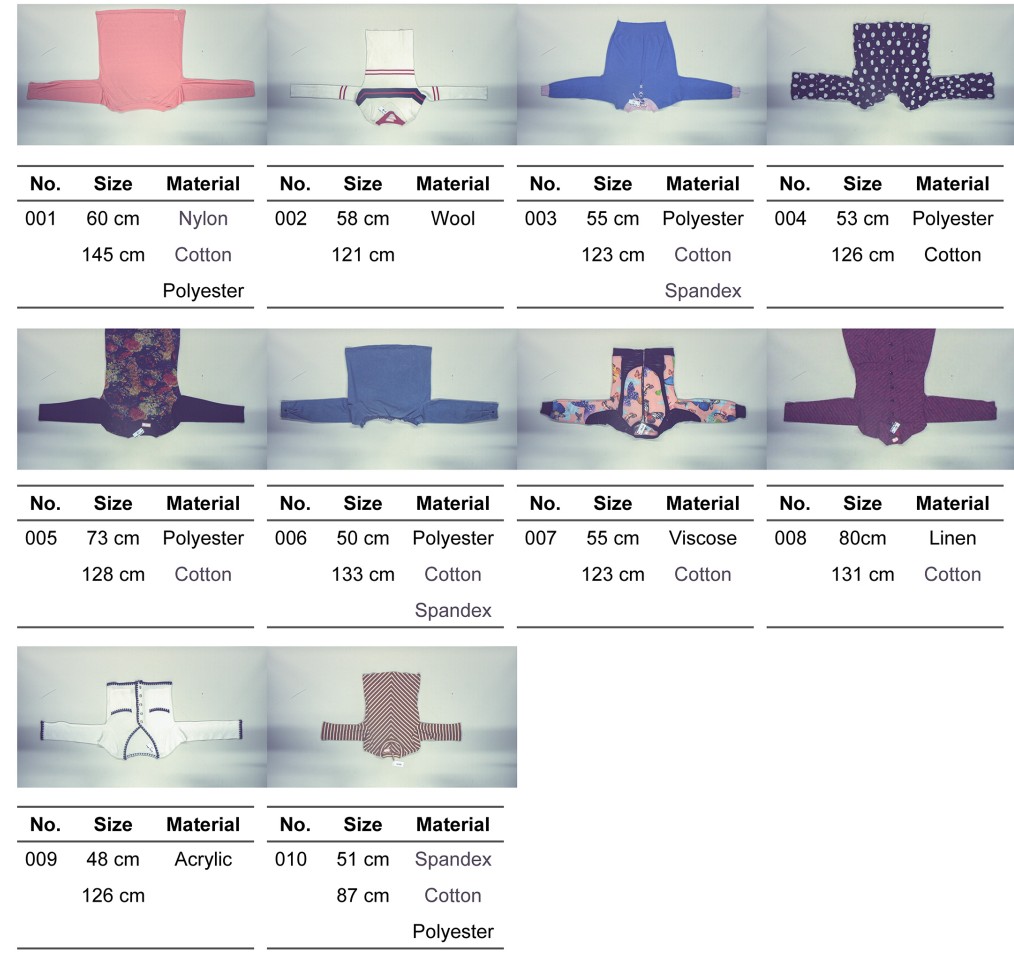

| No. | Size | Material | No. | Size | Material | No. | Size | Material | No. | Size | Material |
|---|---|---|---|---|---|---|---|---|---|---|---|
| 001 | 60 cm | Nylon | 002 | 58 cm | Wool | 003 | 55 cm | Polyester | 004 | 53 cm | Polyester |
|  | 145 cm | Cotton |  | 121 cm |  |  | 123 cm | Cotton |  | 126 cm | Cotton |
|  |  | Polyester |  |  |  |  |  | Spandex |  |  |  |

| No. | Size | Material | No. | Size | Material | No. | Size | Material | No. | Size | Material |
|---|---|---|---|---|---|---|---|---|---|---|---|
| 005 | 73 cm | Polyester | 006 | 50 cm | Polyester | 007 | 55 cm | Viscose | 008 | 80cm | Linen |
|  | 128 cm | Cotton |  | 133 cm | Cotton |  | 123 cm | Cotton |  | 131 cm | Cotton |
|  |  |  |  |  | Spandex |  |  |  |  |  |  |

| No. | Size | Material | No. | Size | Material |
|---|---|---|---|---|---|
| 009 | 48 cm | Acrylic | 010 | 51 cm | Spandex |
|  | 126 cm |  |  | 87 cm | Cotton |
|  |  |  |  |  | Polyester |

Figure 12: Long-sleeve Shirts (Test Set)

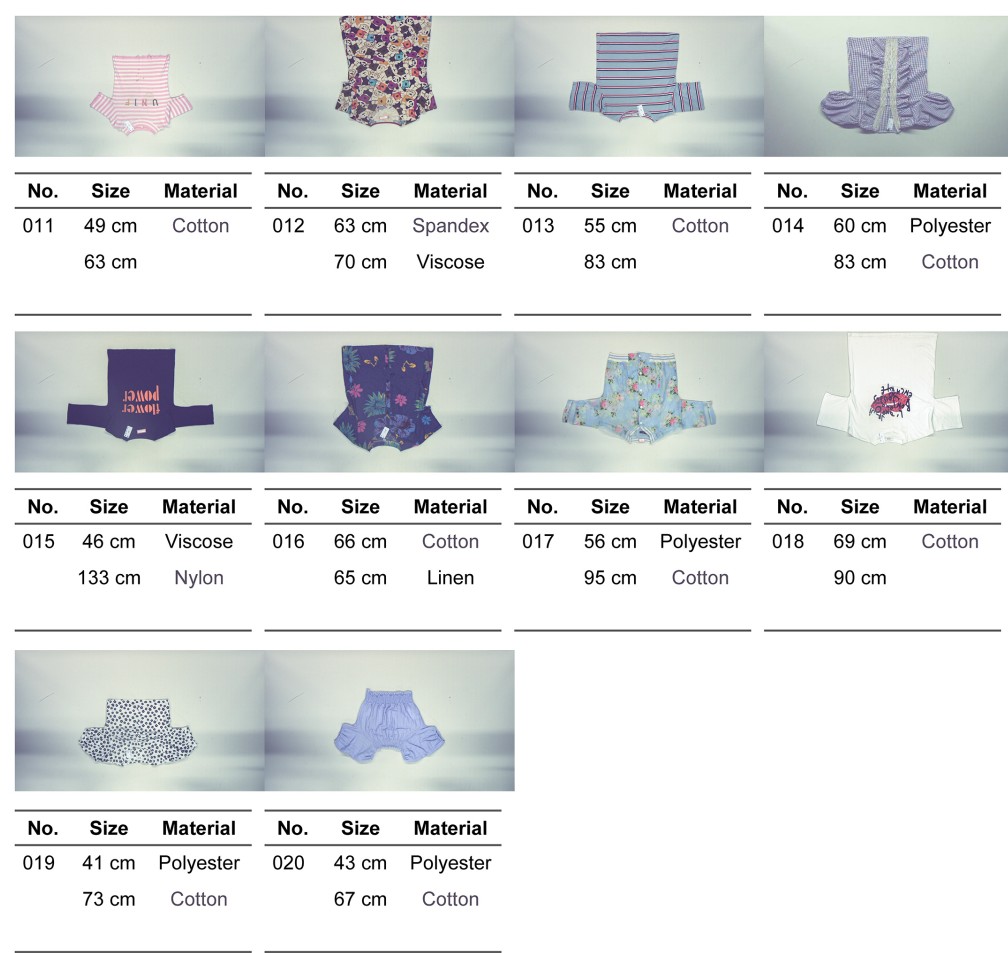

| No. | Size | Material | No. | Size | Material | No. | Size | Material | No. | Size | Material |
|-----|------|----------|-----|------|----------|-----|------|----------|-----|------|----------|
| 011 | 49 cm | Cotton | 012 | 63 cm | Spandex | 013 | 55 cm | Cotton | 014 | 60 cm | Polyester |
|  | 63 cm |  |  | 70 cm | Viscose |  | 83 cm |  |  | 83 cm | Cotton |

| No. | Size | Material | No. | Size | Material | No. | Size | Material | No. | Size | Material |
|-----|------|----------|-----|------|----------|-----|------|----------|-----|------|----------|
| 015 | 46 cm | Viscose | 016 | 66 cm | Cotton | 017 | 56 cm | Polyester | 018 | 69 cm | Cotton |
|  | 133 cm | Nylon |  | 65 cm | Linen |  | 95 cm | Cotton |  | 90 cm |  |

| No. | Size | Material | No. | Size | Material |
|-----|------|----------|-----|------|----------|
| 019 | 41 cm | Polyester | 020 | 43 cm | Polyester |
|  | 73 cm | Cotton |  | 67 cm | Cotton |

Figure 13: Short-sleeve T-Shirts (Test Set)

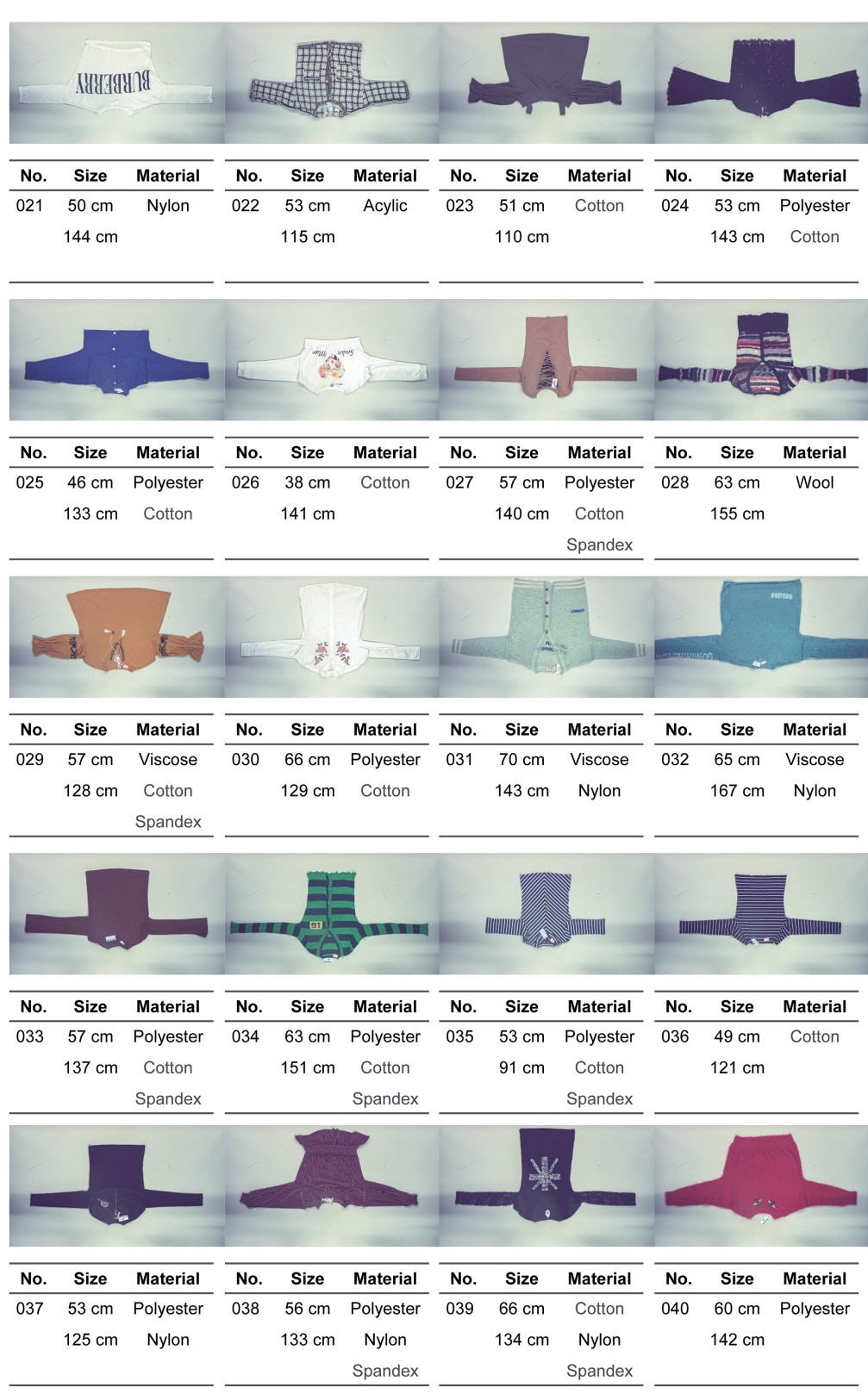

| No. | Size | Material | No. | Size | Material | No. | Size | Material | No. | Size | Material |
|-----|------|----------|-----|------|----------|-----|------|----------|-----|------|----------|
| 021 | 50 cm | Nylon | 022 | 53 cm | Acylic | 023 | 51 cm | Cotton | 024 | 53 cm | Polyester |
| | 144 cm | | | 115 cm | | | 110 cm | | | 143 cm | Cotton |

| No. | Size | Material | No. | Size | Material | No. | Size | Material | No. | Size | Material |
|-----|------|----------|-----|------|----------|-----|------|----------|-----|------|----------|
| 025 | 46 cm | Polyester | 026 | 38 cm | Cotton | 027 | 57 cm | Polyester | 028 | 63 cm | Wool |
| | 133 cm | Cotton | | 141 cm | | | 140 cm | Cotton | | 155 cm | |
| | | | | | | | | Spandex | | | |

| No. | Size | Material | No. | Size | Material | No. | Size | Material | No. | Size | Material |
|-----|------|----------|-----|------|----------|-----|------|----------|-----|------|----------|
| 029 | 57 cm | Viscose | 030 | 66 cm | Polyester | 031 | 70 cm | Viscose | 032 | 65 cm | Viscose |
| | 128 cm | Cotton | | 129 cm | Cotton | | 143 cm | Nylon | | 167 cm | Nylon |
| | | Spandex | | | | | | | | | |

| No. | Size | Material | No. | Size | Material | No. | Size | Material | No. | Size | Material |
|-----|------|----------|-----|------|----------|-----|------|----------|-----|------|----------|
| 033 | 57 cm | Polyester | 034 | 63 cm | Polyester | 035 | 53 cm | Polyester | 036 | 49 cm | Cotton |
| | 137 cm | Cotton | | 151 cm | Cotton | | 91 cm | Cotton | | 121 cm | |
| | | Spandex | | | Spandex | | | Spandex | | | |

| No. | Size | Material | No. | Size | Material | No. | Size | Material | No. | Size | Material |
|-----|------|----------|-----|------|----------|-----|------|----------|-----|------|----------|
| 037 | 53 cm | Polyester | 038 | 56 cm | Polyester | 039 | 66 cm | Cotton | 040 | 60 cm | Polyester |
| | 125 cm | Nylon | | 133 cm | Nylon | | 134 cm | Nylon | | 142 cm | |
| | | | | | Spandex | | | Spandex | | | |

Figure 14: Long-sleeve Shirts (Train Set)

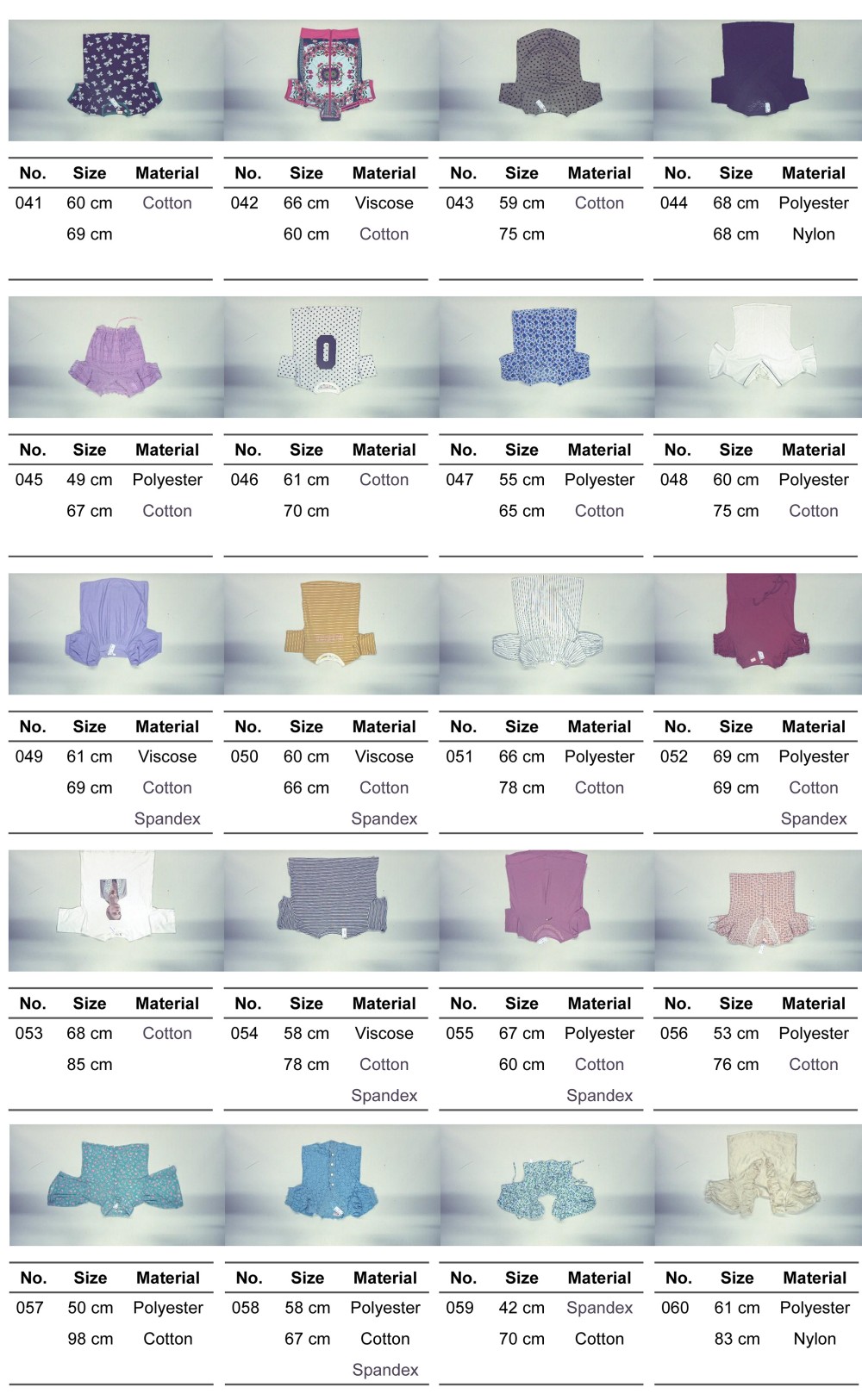

| No. | Size | Material | No. | Size | Material | No. | Size | Material | No. | Size | Material |
|-----|------|----------|-----|------|----------|-----|------|----------|-----|------|----------|
| 041 | 60 cm | Cotton | 042 | 66 cm | Viscose | 043 | 59 cm | Cotton | 044 | 68 cm | Polyester |
|     | 69 cm |        |     | 60 cm | Cotton   |     | 75 cm |        |     | 68 cm | Nylon |

| No. | Size | Material | No. | Size | Material | No. | Size | Material | No. | Size | Material |
|-----|------|----------|-----|------|----------|-----|------|----------|-----|------|----------|
| 045 | 49 cm | Polyester | 046 | 61 cm | Cotton | 047 | 55 cm | Polyester | 048 | 60 cm | Polyester |
|     | 67 cm | Cotton    |     | 70 cm |        |     | 65 cm | Cotton    |     | 75 cm | Cotton |

| No. | Size | Material | No. | Size | Material | No. | Size | Material | No. | Size | Material |
|-----|------|----------|-----|------|----------|-----|------|----------|-----|------|----------|
| 049 | 61 cm | Viscose | 050 | 60 cm | Viscose | 051 | 66 cm | Polyester | 052 | 69 cm | Polyester |
|     | 69 cm | Cotton  |     | 66 cm | Cotton  |     | 78 cm | Cotton    |     | 69 cm | Cotton |
|     |       | Spandex |     |       | Spandex |     |       |           |     |       | Spandex |

| No. | Size | Material | No. | Size | Material | No. | Size | Material | No. | Size | Material |
|-----|------|----------|-----|------|----------|-----|------|----------|-----|------|----------|
| 053 | 68 cm | Cotton | 054 | 58 cm | Viscose | 055 | 67 cm | Polyester | 056 | 53 cm | Polyester |
|     | 85 cm |        |     | 78 cm | Cotton  |     | 60 cm | Cotton    |     | 76 cm | Cotton |
|     |       |        |     |       | Spandex |     |       | Spandex   |     |       |        |

| No. | Size | Material | No. | Size | Material | No. | Size | Material | No. | Size | Material |
|-----|------|----------|-----|------|----------|-----|------|----------|-----|------|----------|
| 057 | 50 cm | Polyester | 058 | 58 cm | Polyester | 059 | 42 cm | Spandex | 060 | 61 cm | Polyester |
|     | 98 cm | Cotton    |     | 67 cm | Cotton    |     | 70 cm | Cotton  |     | 83 cm | Nylon |
|     |       |           |     |       | Spandex   |     |       |         |     |       |        |

Figure 15: Short-sleeve T-Shirts (Train Set)

