# OpenReview forum: "UniFolding: Towards Sample-efficient, Scalable, and Generalizable Robotic Garment Folding"
_robot-learning.org/CoRL/2023/Conference — CoRL 2023 Poster_

### Official Review · Reviewer_Aqt2 · 2023-07-03

**Confidence:** 4
**Originality:** Good
**Technical Quality:** Good
**Clarity Of Presentation:** Very Good
**Impact:** 3

**Recommendation:**

Weak Accept: I recommend accepting the paper, but will not argue for my recommendation if the majority of other reviewers have a different opinion.

**Review:**

Strengths:
* This paper presents a novel unified robotic folding system that supports unfolding and folding, using a single policy.
* Extensive, convincing hardware experiments are shown.
* The paper is well-structured and written.
* Figures are clear and helpful in understanding the method and results.

Weaknesses:
* The paper would greatly benefit from a comprehensive state-of-the-art comparison with well-known robotic folding systems, such as those mentioned in [1] and [2]. While the reproduction of other folding systems might present challenges, including at least one comparison with a widely recognized robotic folding system would significantly enhance the paper's contribution.

**Quality Of The Limitations Section:**

Limitations are addressed clearly

**Questions For Rebuttal:**

Including a state-of-the-art comparison with other well-known robotic folding systems will undoubtedly make the paper stronger. To further strengthen the paper, the following important pieces of information could be included:

1. How does the model perform on the seen garments? Does the model struggle with generalization to unseen garments or learning to unfold/fold in general?
2. How good are the human demonstrations? Do they have perfect, IoU, coverage, and folding success?
3. More details could be provided in the “self-supervised training for unfolding” section. What does “self-supervised training” mean? Is it using a contrastive loss? Or is it training with unlabeled data?
4. Additional analysis could be provided in the “unfold results” section.

Typos:
* Line 109: “random” to “randomly.”

**Robotics Focus:**

Sufficient demonstration on hardware

**Summary Of Paper:**

This paper introduces an innovative and comprehensive robotic folding system capable of both unfolding and folding tasks. The system utilizes an end-to-end neural network, UFONet, to classify primitive actions (fling, first pick-and-place, second pick-and-place, drag, and mop) and regress keypoint candidates, with a partial point cloud of the garment as input. To support the development of the system, a data collection pipeline is proposed: offline data collection in simulation with human demonstrations and online data collection in the real world with humans-in-the-loop. The effectiveness of the system is evaluated through extensive hardware experiments, using both long-sleeved and short-sleeve shirts.


**Summary Of Recommendation:**

I recommend weak accept based on the issues discussed above. The primary concern lies in the need for a state-of-the-art comparison with well-known robotic folding systems. Based on the presented experimental results, judging the quality and performance of the proposed robotic folding system is difficult.

**References:**

[1] Avigal et al., Speedfolding: Learning efficient bimanual folding of garments, IROS 2022.

[2] Ha et al., Flingbot: The unreasonable effectiveness of dynamic manipulation for cloth unfolding, CoRL 2021.

---

### Official Review · Reviewer_3fbS · 2023-07-11

**Confidence:** 4
**Originality:** Fair
**Technical Quality:** Fair
**Clarity Of Presentation:** Good
**Impact:** 3

**Recommendation:**

Weak Accept: I recommend accepting the paper, but will not argue for my recommendation if the majority of other reviewers have a different opinion.

**Review:**

Strengths:
-  The paper introduces a end-to-end system for automatic garments folding. It leverages human demonstration and simulation to improve sample efficiency. It also simplify the actio space into a set of keypoints so that human can label them easily during real world fine-tuning.
- The proposed method is well executed and extensive experiments have been conducted on garments of various shapes and sizes. I really like the folding experiments on long-sleeve shirts.

Weeknesses:
- The claimed advantages over prior works are not well supported by experiments. The paper is largely based off prior works ClothFunnels [1] and SpeedFolding [2], and the main differences are: 1. [1, 2] adopt a heuristic policy for folding while UniFolding learns from demonstration; 2. [1, 2] treats folding and unfolding separately while the proposed method unify both into a single policy. However, I'm not fully convinced that these changes will necessarily lead to performance gains. For example,  SpeedFolding unfolds the garments until it's "sufficiently smooth" before folding. In this case, a simple heuristic policy might be enough. Actually, when looking at the video, I think start folding before the clothes is sufficiently smooth will lead to suboptimal results ( such as 2:29 ).

- Though I appreciate the effort the authors devoted into this work, I have some concerns regarding the scalability of the method. Unlike prior works, the proposed method requires human demonstration and feedback in both simulation and real world. I'm not against using human supervision since it may improve sample efficiency. But I would be interested to see a trade off between human time and robot time. For example, compared to self-supervised training, how much robot training time can be reduced per hour of human time.

**Quality Of The Limitations Section:**

Limitations are addressed clearly

**Questions For Rebuttal:**

- As above, I would like to see more evidence in the advantages of the proposed method. For example, does it improve the efficiency with a unified system? Does it improve accuracy with a learned policy? A side-by-side comparison with prior approaches, either in simulation or real world will be really helpful.
- How does the self-supervised training works? Since the model is supposed to do both unfolding and folding, will self-supervised training on unfolding hurt to performance on folding?

**Robotics Focus:**

Sufficient demonstration on hardware

**Summary Of Paper:**

The paper introduces a robotic system for folding diverse garments, UniFolding. The authors identify 3 challenges for generalizable folding system: long horizon, corner cases and real-world data. To address these challenges, the authors propose an end-to-end neural network, UFONet, which operates using a set of predefined action primitives. There're three contribution highlighted in the papers:
1. A robotic folding system unifies the decision process of unfolding and folding
2. A value-based end-to-end policy that choose the right action types and actioning points for folding
3. Extensive real robot experiments on ~20 garments.

**Summary Of Recommendation:**

The paper proposes a end-to-end system for garment folding and demonstrate it in real world. Overall, it's well executed but more evidence needs to be provided before the paper is accepted.
After rebuttal, I would like to raise my recommendation to weak accept. Although I'm not fully convinced by the novelty and scalability of the method, I think this could be a decent system paper.

---

### Official Review · Reviewer_BruB · 2023-07-14

**Confidence:** 3
**Originality:** Good
**Technical Quality:** Good
**Clarity Of Presentation:** Good
**Impact:** 3

**Recommendation:**

Weak Accept: I recommend accepting the paper, but will not argue for my recommendation if the majority of other reviewers have a different opinion.

**Review:**

Strengths

- This work designs a full pipeline for garment folding that can be deployed in the real world.
- This work shows significant efforts to collect human demonstrations both in simulation and in the real world, which makes training the end-to-end policy network. This methodology may guide future works in garment manipulation.

Weaknesses

- This work does not compare with any previous methods.
   It claims that recent learning methods for garment folding struggles to handle wide variety of garments, and heuristic rules in previous works cannot cover unexpected corner cases. However, it does not justify these claims through experiments.
- Limited technical novelty.
   While it does take non-trivial efforts to create the full garment folding pipeline, the individual modules it uses are not new. The primitive actions, such as fling and pick-and-place, are all introduced in previous works. Keypoint prediction is also already extensively used (e.g., in Speedfolding).
- This paper contains frequent grammar mistakes, but is overall understandable.

**Quality Of The Limitations Section:**

Limitations are addressed clearly

**Questions For Rebuttal:**

The most significant issue of this work is the lack of baseline comparison. Currently, it is not convincing that the proposed method can handle corner cases where heuristic rules cannot, especially when the limitation sections shows many unhandled corner cases. I suggest that this work needs to include comparison with at least a learning based method (e.g. Unifolding[2]) or a heuristic based method. Alternatively, it needs to explain why such comparison is not feasible.

**Robotics Focus:**

Sufficient demonstration on hardware

**Summary Of Paper:**

This work addresses the problem of robotic garment folding. It supports the full pipeline including unfolding and folding.
It proposes an end-to-end policy model, UFONet, to make decisions through the folding procedure. UFONet takes a partial camera point cloud as input, classifies the current action primitive to execute, and regresses the action points. The action primitives are achieved with controllers adapted from previous works (fling), or rule-based control (pick-and-place, drag & mop).
UFONet is first pre-trained with supervised learning using human demonstrations collected with VR, then trained in a self-supervised setting in simulation. Finally, it is fine-tuned in the real world where humans annotate data collected from deploying the policy in the real world.

**Summary Of Recommendation:**

While I am not currently convinced that the proposed method would have significant advantage over previous methods such as Unifolding due to the lack of comparison. The method itself seems correct, takes significant efforts to implement, and may inspire future research, thus I lean towards accept.

---

### Official Review · Reviewer_Pcsh · 2023-07-17

**Confidence:** 4
**Originality:** Good
**Technical Quality:** Good
**Clarity Of Presentation:** Good
**Impact:** 2

**Recommendation:**

Weak Reject: I recommend rejecting the paper, but will not argue for my recommendation if the majority of other reviewers have a different opinion.

**Review:**

The paper is well-written and easy to understand. The system proposed (UniFolding) is a complete pipeline for folding garments (for now, shirts). Occasionally the system will need to unfold in order to make progress on the overall folding task - this is handled organically by the system (a noteworthy aspect of the paper). Another positive in the work is the set of results on real shirts. On the less positive side, the paper is really not about brand-new ideas, but this is not a huge concern (most papers build on others' results) since I think a 'systems' approach to putting together older ideas and testing on real robots is a valid way to proceed. The system evaluation is lacking, however. For this, please see my remarks on rebuttal below.





**Quality Of The Limitations Section:**

Limitations are addressed clearly

**Questions For Rebuttal:**

It's not clear that one can take away anything of significance from the row labeled Folding Success in Table 1. I understand these are average success rates as scored by three humans, but this doesn't seem like a reliable measure (did the humans just see the final folded shirt, or the whole process ? who were the human evaluators? did you have them score a human folded shirt or one folded by a small child to get some sense of what they considered success ? etc). Also, I don't understand the paragraph labeled Success Rate for End-to-end Folding on page 7. Specifically, where it says 'crumpled initial state' do you mean 'crumpled final state' ? I'm also not sure how crumpled is defined.

**Robotics Focus:**

Sufficient demonstration on hardware

**Summary Of Paper:**

At its heart, this is a systems paper for robot folding of garments (specifically, shirts). There is some diversity of garments (short vs long sleeves, texture, shape, etc), and the results are shown on real shirts. The method treats folding and unfolding via a unified policy model and works on a partial point cloud,

**Summary Of Recommendation:**

Overall, a nice paper, but it could do with additional rigor in evaluation.

---

### Author Response · Authors · 2023-08-16
**Response to All Reviewers**

We thank all the reviewers for the thoughtful reviews. We have added extra supplementary files for rebuttal based on the reviewer's feedback and addressed their questions below. Here is the summary:
1. Added experiment results of **baseline comparison on folding policy** (learned v.s. heuristic).
2. Added experiment results of **system-level comparison with a state-of-the-art baseline**.
3. Added experiment results on **sample-efficiency and scalability** (as suggested by reviewer 3fbS).
4. Added experiment results on the training set (as suggested by reviewer Aqt2).
5. Provided clarification on techinal novelty (as suggested by reviewer BruB).
6. Provided clarification on evaluation methods (as suggested by reviewer Pcsh).
7. Provided clarification on self-supversied learning.

---

### Decision · Program_Chairs · 2023-08-30

**Decision:**

Accept (Poster)

**Comment:**

This paper presents a system for cloth folding. The original reviews are mixed with the major concern about the technical novelty compare to existing methods such as ClothFunnels and SpeedFold. In the rebuttal, the author clarified and pointed out some system-level comparisons with the existing method, which addresses some of the major concerns.

While the AC agree that the paper does not present many novel idea or method, we can also see the value of the systems approach that puts together exsiting methods to compose a new and effective system and evaluate it on a real robot system. This excise could provide valuable information for future work in this area and get the methods one step closer to an effective real-world cloth folding system.  Based on this evaluation, AC recommends accepting this paper. The author is encouraged to include the additional result and discussion based on reviewer feedback.